DATA RELEASE

# A genome assembly of the Atlantic chub mackerel (*Scomber colias*): a valuable teleost fishing resource

André M. Machado[1,2], André Gomes-dos-Santos[1,2], Miguel M. Fonseca[1], Rute R. da Fonseca[3,4], Ana Veríssimo[5,6], Mónica Felício[7], Ricardo Capela[1,2], Nélson Alves[1,2], Miguel Santos[1,2], Filipe Salvador-Caramelo[1,2], Marcos Domingues[1], Raquel Ruivo[1], Elsa Froufe[1] and L. Filipe C. Castro[1,2,*]

1 CIIMAR – Interdisciplinary Centre of Marine and Environmental Research, U. Porto – University of Porto, Porto, Portugal
2 Department of Biology, Faculty of Sciences, U. Porto - University of Porto, Portugal
3 Center for Global Mountain Biodiversity, GLOBE Institute, University of Copenhagen, Copenhagen, Denmark
4 Center for Macroecology, Evolution, and Climate, GLOBE Institute, University of Copenhagen, Denmark
5 CIBIO - Centro de Investigação em Biodiversidade e Recursos Genéticos, InBIO - Laboratório Associado, Campus de Vairão, Universidade do Porto, 4485-661 Vairão, Portugal
6 BIOPOLIS - Program in Genomics, Biodiversity and Land Planning, CIBIO, Campus de Vairão, 4485-661 Vairão, Portugal
7 Portuguese Institute for the Sea and Atmosphere, I.P. (IPMA), Portugal

## ABSTRACT

The Atlantic chub mackerel, *Scomber colias* (Gmelin, 1789), is a medium-sized pelagic fish with substantial importance in the fisheries of the Atlantic Ocean and the Mediterranean Sea. Over the past decade, this species has gained special relevance, being one of the main targets of pelagic fisheries in the NE Atlantic. Here, we sequenced and annotated the first high-quality draft genome assembly of *S. colias*, produced with PacBio HiFi long reads and Illumina paired-end short reads. The estimated genome size is 814 Mbp, distributed into 2,028 scaffolds and 2,093 contigs with an N50 length of 4.19 and 3.34 Mbp, respectively. We annotated 27,675 protein-coding genes and the BUSCO analyses indicated high completeness, with 97.3% of the single-copy orthologs in the Actinopterygii library profile. The present genome assembly represents a valuable resource to address the biology and management of this relevant fishery. Finally, this genome assembly ranks fourth in high-quality genome assemblies within the order Scombriformes and first in the genus *Scomber*.

**Subjects** Genetics and Genomics, Evolutionary Biology, Marine Biology

**Submitted:** 22 November 2021

* Corresponding author. E-mail: lfilipecastro@gmail.com; filipe.castro@ciimar.up.pt

Preprint submitted at https://doi.org/10.1101/2021.11.19.468211

## DATA DESCRIPTION

### Background and context

The family Scombridae is divided into 2 subfamilies (Gasterochismatinae and Scombrinae), with 15 genera and around 49 described species, comprising mackerels, bonitos, and tunas [1]. The representative genus of the Scombridae, i.e., *Scomber*, includes 4 species: *S. scombrus*, *S. japonicus*, *S. australasicus*, and *S. colias*. The Atlantic chub mackerel, *Scomber colias* (Gmelin, 1789) (NCBI:txid338315, FishBase ID:54736), is a small coastal pelagic fish that is distributed widely, being found in the Atlantic Ocean from the Bay of

**Figure 1.** Photograph of Atlantic chub mackerel, *Scomber colias*. The specimen was caught in 2020 and used for Pacbio HiFi genome assembly.

Biscay to South Africa (including the Canary, Madeira, Azores, and Saint Helena Islands), and in the Mediterranean Sea (Figure 1) [2]. *Scomber colias* is usually found at depths of up to 300 m and occupies a key position in the trophic web. This species acts as a link between primary producers and top predators, since it feeds mainly on zooplankton and small pelagic fish, and is an essential element of the diet of larger pelagic fish (e.g., tuna, swordfish, and sharks) and marine mammals (e.g., dolphins and seals) [3]. Besides its ecological importance, *S. colias* also supports important commercial fisheries for several countries across its distribution range, being an important component in the diet of local populations [1, 4]. This is probably related to its nutritional value, as this mackerel is a valued source of important fatty acids for human nutrition, particularly docosahexaenoic acid (DHA), an omega-3 fatty acid [5, 6]. Additionally, *S. colias* is used as bait for the tuna longline and handline fisheries, and is caught in purse seine and pelagic trawl fisheries which target sardines and anchovies [7].

The availability of *S. colias* makes it a sustainable marine resource [6] and a viable alternative to the European sardine (*Sardina pilchardus*), which is under fishing restrictions due to a population decline. Curiously, fluctuations in abundance and a northwards shift in the distribution of *S. colias*, with a likely inverse relationship with sardine abundance, have been recently demonstrated [8]. Due to its ecological and economic importance, *S. colias* has been the focus of several recent studies on different aspects of its fisheries and biology [3, 8, 9]. Yet, genomic resources for the species are still limited. Presently a (liver) transcriptome [10], a mitogenome [11], and single-nucleotide polymorphism (SNP) data obtained through restriction site-associated DNA sequencing [12], have been described for the species. With the vast majority of the world's fish stocks already in collapse, and with climate change as additional pressure, information on fish genomes is becoming a pressing tool to address conservation efforts [13, 14]. Here, we report the first high-quality draft genome of *S. colias*, assembled with Illumina and Pacific Biosciences (PacBio) Single Molecule High-Fidelity (HiFi) reads. This resource provides a critical platform to uncover the species' adaptive physiological potential in a changing environment. Specifically, it will help understand the current observed populational northward shift, postulated to be part of a more general expansion of species from warmer areas [8]. Moreover, being one of the genomes with higher quality within the family Scombridae and the first within the *Scomber*

genus, this information will help to improve the conservation, management, and sustainable exploitation of this valuable fish resource as well as that of its highly valued congeners.

## METHODS

### Sampling and DNA extraction

Two specimens of *S. colias* were collected at 2 sampling time points. The first specimen was collected in 2017, during the "*Programa Nacional de Amostragem Biológica*" managed by the Instituto Português do Mar e da Atmosfera" (IPMA), in North Atlantic waters (41.501944 N 8.851667 W). From this individual, 2 tissue types were collected, and were stored in 100% ethanol (muscle) or RNA *later* (liver). Liver tissue was used to produce and describe the first liver transcriptome of *S. colias* [10]. Muscle tissue was used in the present study, for genomic DNA (gDNA) extraction using the DNeasy Blood and Tissue Kit (Qiagen, Hilden, Germany), following the manufacturer's instructions. The gDNA was then used for Illumina paired-end (PE) sequencing (described below). The second specimen was caught in 2020, near Mira, Portugal (40.5588270 N 9.4529720 W). Immediately upon harvesting, the muscle was snap frozen in liquid nitrogen. The frozen tissue was shipped to Brigham Young University DNA Sequencing Center (BYU), where gDNA with high molecular weight was extracted from 1.1 g of muscle using the QIAGEN Genomic-tip 20/G kit. The quality and concentration of the gDNA were assessed with Qubit Fluorometric system (ThermoFisher), and the fragment size was determined with a fragment analyser (Agilent Technologies, RRID:SCR_013575) before loading on the Pacbio Sequel II system (PacBio Sequel II System, RRID:SCR_017990).

### DNA sequencing libraries construction and sequencing

For the first DNA sample, Illumina PE library preparation and sequencing were carried out by Macrogen, Inc. (Seoul, Korea), using Illumina HiSeq X Ten platform (Illumina HiSeq X Ten, RRID:SCR_016385), with 250 bp PE configuration. For the second specimen, PacBio HiFi library preparation and sequencing were performed at BYU, following the manufacturer's recommendations [15]. The size-selected fraction had a mean read length of 15.3 kbp and was selected on the SageELF system (Sage Science, RRID:SCR_014808). The sequencing was conducted on two single-molecule, real-time (SMRT) cells using Sequel II system v.9.0, with a run time of 30 h, and 2.9 h pre-extension. The circular consensus analysis was performed in SMRT® Link v9.0 [16] under default settings (the statistics of raw data generated from each PacBio SMRT cell can be viewed in Additional File 1 [17, 18]).

### Raw data quality control, clean-up, and genome size estimation

Both short- and long-read datasets were assessed by FastQC v.0.11.8 software (FastQC, RRID:SCR_014583). Trimmomatic v.0.38 software (Trimmomatic, RRID:SCR_011848) [19] was used to filter and remove low quality reads as well as the adaptors of the Illumina dataset (LEADING:5 TRAILING:5 SLIDINGWINDOW:4:20 MINLEN:50). Next, trimmed datasets were used to check the overall characteristics of the *S. colias* genome (i.e., genome size, heterozygosity, unique content), through GenomeScope 2.0 [20]. Briefly, Jellyfish v.2.2.10 software (Jellyfish, RRID:SCR_005491) [21] was used to build *k*-mer frequency distributions, and the final *k*-mer counts (*k*-mer 21, 25, 31) were submitted to the GenomeScope 2.0 online platform. On the other hand, HiFi reads were filtered in two ways (Figure 2). First, mitochondrial reads were removed by BLAST searches (BLASTN, RRID:SCR_001598) using a

prebuilt database of mitochondrial sequences (database build protocol: (1) select all complete mitogenomes present in the nucleotide database of the National Center for Biotechnology Information (NT-NCBI); (2) select by taxon (Actinopterygii; txid:7898); (3) sequence length filter 15,000–50,000 bp; (4) build a database with the makeblastdb application of NCBI-BLAST+ v.2.9.0). Second, to filter out possible sources of contamination (artefactual or biological), HiFi reads were checked by BLAST (BLASTN) against NT-NCBI. Only HiFi reads with match hits over 90% identity and query coverage of 50% in the Actinopterygii taxon (NCBI:txid7898), or without match hits at all, were considered for further analysis (Figure 2).

## Mitochondrial genome assembly

Given that 2 specimens were used for the distinct sequencing approaches, i.e., PacBio HiFi and Illumina PE, the whole mitochondrial genome (mtDNA) was assembled and characterised for both specimens. For specimen 1, trimmed Illumina PE reads were used to assemble mtDNA in GetOrganelle v.1.7.1 [22] with optimised parameters (-F animal_mt -w 121 -R 10 -k 85,95,105,115,125) (Figure 2). For specimen 2, a new pipeline was designed to produce a mtDNA assembly from the PacBio HiFi long reads (Figure 2). The PacBio HiFi mtDNA reads, previously filtered (see above), were corrected using Hifiasm v.0.13-r308 (Hifiasm, RRID:SCR_021069) [23] with optimised parameters (–write-ec). Since Hifiasm is not optimised to assemble circular molecules (which are expected for mtDNA), the corrected PacBio HiFi mtDNA reads were assembled using Unicycler v.0.4.8 [24], a software package designed to assemble bacterial genomes and so optimised for circular assemblies, with default parameters. Annotation and visual representation of both mtDNA assemblies were produced using MitoZ v.2.3 [25] with optimised parameters (–genetic_code 2; –clade Chordata; –topology circular), using the PE reads for coverage plotting. Furthermore, annotations were manually validated by comparison with other mitochondrial genomes of the genus *Scomber*, available at NCBI (see Data Availability).

## Nuclear genome assembly and assessment

For whole-genome assembly a combined approach, using short- and long-read assemblies, was applied (Figure 2). While long-read assemblies were mainly used to produce the primary assembly, short-read assemblies were used to scaffold and improve the contiguity of the basal assembly. In summary, short-read assemblies were performed with the W2RAP pipeline v.0.1 [26], following the authors' protocol. First the *k*-mer analyses toolkit (KAT) v.2.4.1 software (KAT, RRID:SCR_016741) [27] hist module was applied to determine the ideal *k*-mer cut-off, before W2RAP with optimised parameters (-t 30; -m 500; –min_freq 14; -d 32; –dump_all 1; -k: 144, 180, 200, 224) was used to produce 4 assemblies (Figure 2). To generate the long-read assembly, multiple software and parameters were initially tested. PacBio HiFi reads were assembled in Hifiasm v.0.13-r308 [23] with a range of parameters (*k* = 21, 25, 31, 41, 45, 51; *l* = 0, 2) and in HiCanu v.2.1.1 [28] with optimised parameters (default). While Hifiasm generated 2 pseudo-haplotypes per assembly, HiCanu generated 1 merged assembly. To choose the "best" assembly we applied a series of analyses, including Bandage (a bioinformatics application for nagivating *de novo* assembly graphs easily) v.0.8.1 [29] and manual inspection; Benchmarking Universal Single-Copy Orthologs (BUSCO) v.5.2.2 (BUSCO, RRID:SCR_015008) [30] with Eukaryota and Actinopterygii databases was used to assess the gene completeness of the assemblies, and Quality Assessment Tool for Genome Assemblies

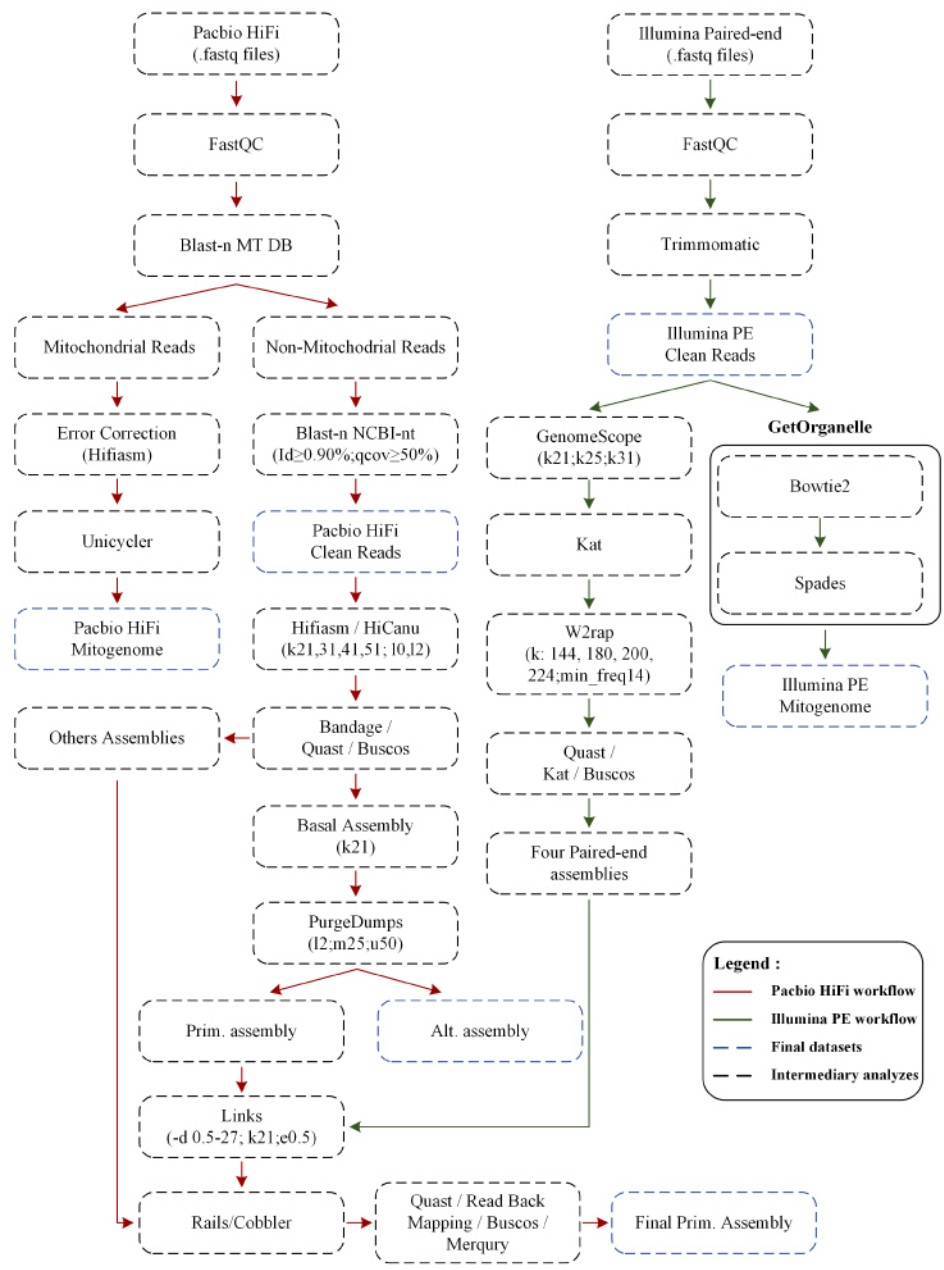

**Figure 2.** Bioinformatics workflow used to perform the genome assembly of *Scomber colias* species.

(QUAST) v.5.0.2 (QUAST, RRID:SCR_001228) [31], to check general metrics of the assemblies (Figure 2). Due to discrepancies in the length of the Hifiasm primary and alternative pseudo-haplotypes, we chose to concatenate them in a single assembly. At this point, the assembly with the highest complete BUSCO scores, highest contiguity (N50), and longest contig, was selected for further analysis. The pseudo-haplotypes were separated by purge_dups v.1.2.5 (purge dups, RRID:SCR_021173) [32]. After the first round of purging and inspection by *k*-mer plot, produced by the KAT tool, cutoffs were manually adjusted.

To assess the influence of purge_dups in the genome, BUSCO (rate of deduplicates) and QUAST (N50 and genomic length per pseudo-haplotype) were used. Next, to improve the contiguity and quality of the assembly, short-read assemblies were used to structurally scaffold the assembly without the introduction of any new bases in the assembly, similar to the literature [33, 34] (Figure 2). The 4 short-read assemblies were inputted to the Long Interval Nucleotide *K*-mer Scaffolder (LINKS) v.1.8.7 [35], being used as long reads; using several distance values, i.e., -d 0.5, 1.5, 3, 9, 27 kb, the primary assembly was rescaffolded interactively for 5 rounds (additional parameters: -k 21 -e 0.5). Furthermore, the scaffolded genome and the long-read assemblies, initially produced by Hifiasm and HiCanu and discarded based on contiguity and completeness, were inputted to Cobbler v.0.6.1 [36] and RAILS v.1.5.1 [36] pipeline, with default parameters. This allowed gap filling of ambiguity regions (produced by short-read scaffolding), and further rescaffolding using long-read information. To evaluate the final assembly, several metrics and software were used. In addition to BUSCO and QUAST metrics, read back mapping of paired-end (PE) reads with Burrows-Wheeler Aligner (BWA) v.0.7.17-r1198 (BWA, RRID:SCR_010910) [37], long reads with Minimap2 v.2.17 (Minimap2, RRID:SCR_018550) [38] and RNA sequencing (RNA-Seq) with Hisat2 v.2.2.0 (HISAT2, RRID:SCR_015530) [39, 40], were also applied. To check consensus quality (QV) and *k*-mer completeness we used Merqury v.1.1 [41] (Figure 2).

## Repeat masking, gene prediction, and annotation

The repetitive elements of the genome were predicted and masked by RepeatMasker v.4.0.7 (RepeatMasker, RRID:SCR_012954) [42] using homologous comparisons and *ab initio* predictions. First, the *de novo* library of repetitive elements was created with the RepeatModeler v.2.0.1 (RepeatModeler, RRID:SCR_015027) [43]. Next, the *ab initio* library, as well as the Dfam_consensus-20170127 (Dfam, RRID:SCR_021168) [44] and RepBase-20181026 (Repbase, RRID:SCR_021169) [45], were used in RepeatMaker to softmask the *S. colias* genome assembly. The genome annotation was performed with the BRAKER2 pipeline v.2.1.6 (BRAKER, RRID:SCR_018964) [46–48]. Initially, the liver RNA-Seq reads (accession number: SRR6367407 [10]) were downloaded, mapped against the *S. colias* genome assembly using Hisat2 v.2.2.0 [39, 40] with default parameters, and converted to BAM and sorted files using Samtools v.1.9 (SAMTOOLS, RRID:SCR_002105) [49]. Additionally, we collected 89 proteomes from NCBI RefSeq (RefSeq, RRID:SCR_003496) [50] and Ensembl (Ensembl, RRID:SCR_002344) [51] databases. The species and accession numbers of the proteomes used in the genome annotation of *S. colias* can be consulted in Additional File 2 [17, 18]. Of these, 82 species belong to the class Actinopterygii (32 taxonomic orders): 81 with genome assembly at chromosome level, and 1 at scaffold level. As of the date of this genome annotation, only 1 Scombriforme genome, *Thunnus orientalis*, was annotated at scaffold level. The remaining 7 proteomes were selected from other vertebrate non-teleost animal models: *Callorhinchus milii*, *Amblyraja radiata*, *Scyliorhinus canicula*, *Lepisosteus oculatus*, *Petromyzon marinus*, *Mus musculus*, and *Homo sapiens*. Next, the RNA-Seq alignment, as well as all the above-mentioned proteomes, were inputted to the BRAKER2 pipeline with optimised parameters (–etpmode; –softmasking; –UTR = off; –crf; –cores = 30). The final file of predictions (braker.gtf) was further filtered by evidence, keeping only gene predictions with RNA-Seq or protein evidence (using BRAKER2 auxiliary scripts; selectSupportedSubsets.py), then converted to .gff3 format (using the Augustus auxiliary scripts; gtf2gff.pl) and post-processed with Another Gtf/Gff Analysis Toolkit (AGAT)

v.0.6.0 [52]. The post-processing stage involved the correction of overlapping gene prediction coordinates and the removal of small or incomplete protein-coding genes (i.e., coding for <100 amino acids (aa); lacking start or stop codons). Furthermore, the proteins were extracted with AGAT and subject to functional annotation using InterProScan v.5.44.80 (InterProScan, RRID:SCR_005829) [53] and BLASTP (BLASTP, RRID:SCR_001010) searches against RefSeq [50] and UniProtKB/SwissProt (UniProtKB, RRID:SCR_004426) [54] databases. The homology searches were performed with DIAMOND v.2.0.11.149 (DIAMOND, RRID:SCR_016071) [55] with optimised parameters (-k 1, -b 10, -e 1e-5, –ultra-sensitive, –outfmt 6). Finally, the genome and the annotation datasets were integrated using JBrowse2 (JBrowse, RRID:SCR_001004) [56], a dynamic web platform for genome visualisation and analysis that allows easy and interactive exploration of provided data (http://portugalfishomics.ciimar.up.pt/app/scombercolias/). The FASTA file containing the genome was indexed with Samtools faidx v.1.9 [49] and added to the JBrowse component, along with the annotation file sorted with GenomeTools v.1.6.1 (GenomeTools, RRID:SCR_016120) [57], and indexed with Samtools tabix v.1.9 [58]. In addition to the JBrowse component, NCBI-BLAST+ v.2.12.0 [59] was integrated into the webpage, allowing BLAST results from the genome, mRNA, protein-coding sequences (CDS), and proteins, directly from the website.

## Phylogenomics

To generate a phylogenomic analysis, the proteomes of 15 selected Actinopterygii species, including the Scombriformes species *Thunnus maccoyii* and *T. orientalis*, were downloaded from public databases. The species and accession numbers used in the phylogenomic analyses can be consulted in Additional File 2 [17, 18]. Single-copy orthologs between these 15 species and *S. colias* were retrieved from the protein datasets by constructing protein family clusters using OrthoFinder v.2.4.0 (OrthoFinder, RRID:SCR_017118) [60] with optimised parameters (-M). This resulted in a total of 392 single-copy orthologous sequences that were individually aligned using MUSCLE v.3.8.31 (MUSCLE, RRID:SCR_011812) [61] with default parameters. Each alignment was trimmed using TrimAl v.1.2 (trimAl, RRID:SCR_017334) [62] with a gap threshold of 0.5 with optimised parameters (-gt 0.5), and afterwards concatenated using FASconCAT-G [63]. Phylogenetic inferences were conducted in IQ-Tree v.1.6.12 (IQ-Tree, RRID:SCR_017254) [64] with optimised parameters (-bb 10000 -nt AUTO -st AA). The best-fit molecular evolutionary model used in the phylogenetic analyses was JTT+F+R4, which was selected by ModelFinder [65] implemented within IQ-Tree.

## Assessing the nuclear receptor and the *"chemical defensome"* repertoire in *Scomber colias*

To demonstrate the value of the present genome resourse, we collected the repertoire of nuclear receptors (NRs) in *S. colias* via TBLASTN (TBLASTN, RRID:SCR_011822) searches in the primary genome assembly with default parameters. Protein sequences of DNA-binding domains and ligand-binding domains in *H. sapiens* NRs were collected from the RefSeq [50] database and used in a query (NP_000466.2, NP_068804.1, NP_003241.2, XP_005257609.1, NP_001349802.1, NP_068370.1, NP_599022.1, NP_009052.4, NP_001351014.1, XP_005260464.1, NP_002948.1, NP_001257330.1, NP_003288.2, XP_016862607.1, NP_001273031.1, NP_005645.1, NP_001278159.1, NP_004442.3, NP_000167.1, XP_005268879.1, NP_004950.2, NP_201591.2).



Next, regions aligning with *H. sapiens* sequences were collected, translated to protein using the Bio.Seq module of Biopython v.1.75 (Biopython, RRID:SCR_007173) [66], and blasted (BLASTP) against a local database containing the NR proteins of *Danio rerio* (*D. rerio* NRs database protocol: (1) NRs sequences and classifications were retrieved from [67]; (2) an NRs database was built using the makeblastdb application of NCBI-BLAST+ v.2.12.0). For each NRs sequence in *S. colias*, the best blast hit in the *D. rerio* database was collected. In some cases, several NRs of *S. colias* matched the same receptor in *D. rerio*. In these cases, the nucleotide sequences of *S. colias* were again validated against the NT-NCBI database, and all sequences matching different GeneIDs in the same organism were kept in the final table of NRs. In parallel, and to assess the genome annotation performed by BRAKER2, the genomic coordinates of regions aligning with *H. sapiens* were searched and identified in the annotation files.

To identify the genes related to the *chemical defensome*, target genes were selected based on a previous report profiling the "chemical defensome" of teleost species [68]. Next, gene names were used as queries to search the deduced *S. colias* genome annotation, a simple but successful approach for well-annotated genomes such as *D. rerio* [68]. When gene names were not retrieved from *S. colias* genome annotation (i.e., *fthl*, *gstp*, *hsph*, *maff*, *nme8*, *slc21*), further TBLASTN searches were performed in the primary genome assembly with optimised parameters (-max_hsps 1 to keep the best query-subject pair), using *D. rerio* sequences as a query.

## Demography with pairwise sequentially Markovian coalescent (PSMC)

To explore the variation in the demographic history of the species, a pairwise sequentially Markovian coalescent (PSMC) (PSMC, RRID:SCR_017229) strategy was applied [69], following the authors' instructions. Briefly, PE short reads were aligned to the repeated masked genome assembly using BWA v.0.7.17-r1198 (BWA, RRID:SCR_010910) [37] with optimised parameters (BWA-MEM), and the output converted to BAM format and sorted using Samtools v.1.9 [49] (function: sort; parameters: default). Next, Picard Tools v.2.19.2 (Picard, RRID:SCR_006525) was used to remove duplicate reads (function: MarkDuplicates; parameters: default), and SAMtools for mapping quality filtering and SNP calling (function: mpileup; parameters: -Q 30 -q 30 -C 50). BCFtools v.1.9 (SAMtools/BCFtools, RRID:SCR_005227) was applied to extract consensus sequences (function: call; parameters: -c), and the subscript vcfutils (from SAMtools) was used for filtering the output for a minimum depth of 25, a maximum depth of 150, and a min RMS mapQ of 20 (function: vcf2fq; parameters: -d 25 -D 150 -Q 20). The resulting fastq file was converted to a PSMC-compatible input format using fq2psmcfa with a minimum quality threshold of 20 (parameters: -q 20). Inferences of population history were performed by running PSMC for 25 iterations with optimised parameters (-N 15, -r 5, -p 4*4 + 13*2 + 4*4 + 6) following recent PSCM estimations on Scombriformes [70]. Furthermore, to account for uncertainties in the PSMC estimates, bootstrapping of 100 replicates was performed using the split face script provided by PSMC authors. Finally, to scale the demographic estimations, a mutation rate (μ) of $7.3 \times 10^{-9}$ substitutions/site/generation was used, based on a recent estimation for the Scombriformes species *Thunnus albacares* [70], and a generation time for *S. colias* of 2 years [7, 71].

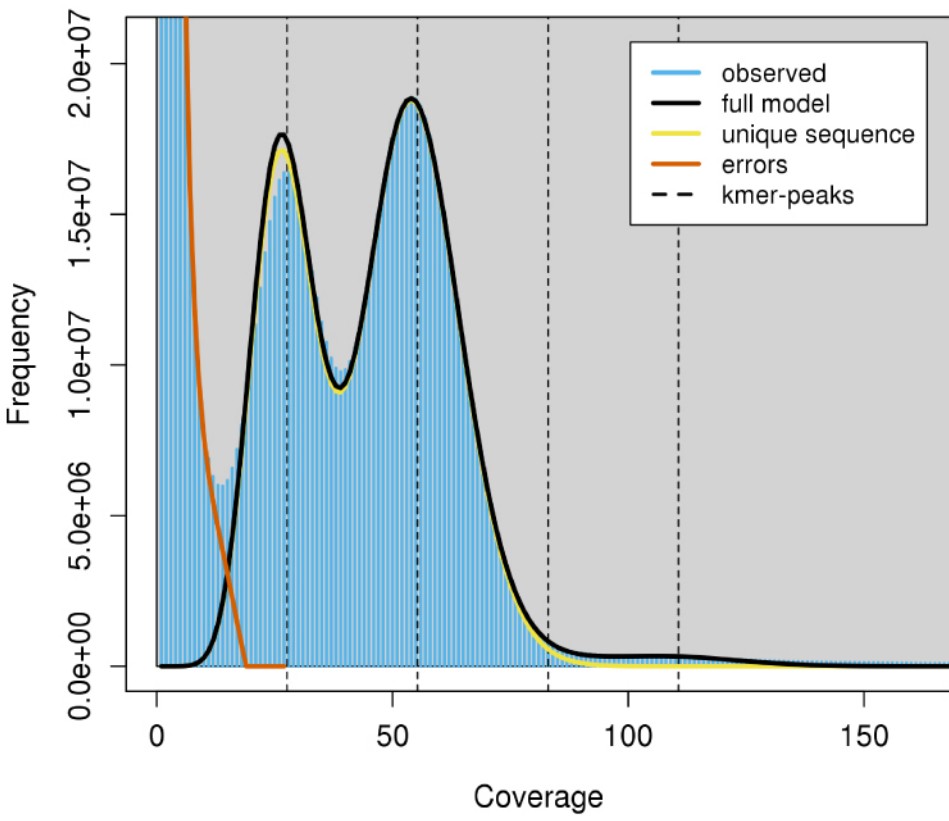

**Figure 3.** Genomescope2 plot with *k*-mer spectra content and fitted models of the *Scomber colias* Illumina PE dataset.

## DATA VALIDATION

To produce the *S. colias* genome assembly, 2 sequencing strategies were used: Illumina PE short reads and PacBio HiFi long reads. The PE dataset was used to assess the genomic proprieties of the *S. colias* species and scaffold the long-read assembly, while HiFi reads were used to perform the primary genome assembly and gap closing (Figure 2).

The Illumina sequencing yielded 149 M of PE reads and the PacBio sequencing generated 1.7 M of HiFi reads (Table 1). Trimmed short reads were used to estimate the genome size (817 Mbp), heterozygosity rate (1.31%), and genome repeat content (approximately 26%), using GenomeScope2 (Figure 3). The complete statistics of GenomeScope2 can be consulted in Additional File 3 [17, 18]. In parallel, the HiFi dataset was inspected, and mitochondrial reads, as well as possible sources of contamination, were removed (amounting to 0.31% of the initial dataset) (Table 1).

For the mtDNA assemblies, a total of 38,868 mtDNA PE reads were filtered by GetOrganelle and a total of 792 mtDNA PacBio HiFi reads were filtered by BLASTN search. The 2 assemblies had the same length, 16,570 bp, and differed from each other by 0.29%

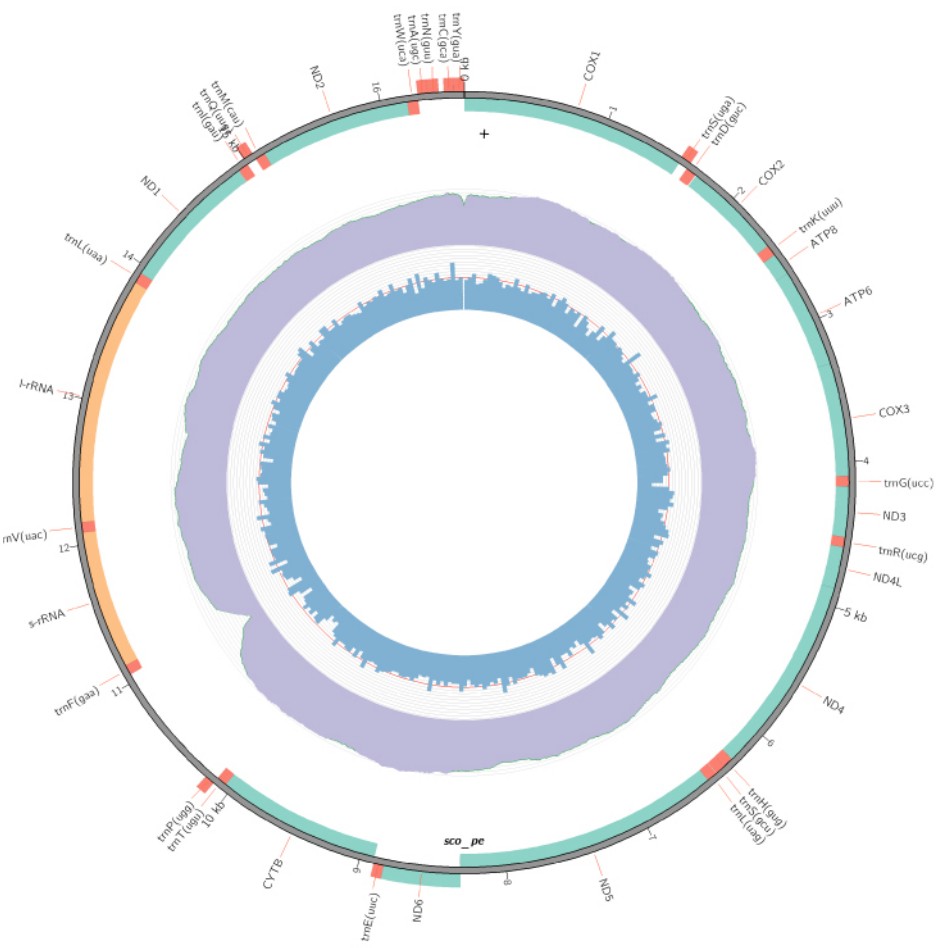

**Figure 4.** Circular mitochondrial genome assembly of *Scomber colias*, obtained from the Illumina PE dataset (equal to that obtained from the PacBio HiFi long reads assembly). From the centre to the outmost features: GC content distribution; sequencing depth distribution of aligned Paired-End reads; gene elements (i.e., PCGs, rRNA genes, tRNA genes).

**Table 1.** General statistics of read datasets used to perform the *Scomber colias* genome assembly.

| Sample | Sequencing type | Library type | Platform | Insert size (bp) | Number of reads (before clean-up) | Number of reads (after clean-up) | Application |
|---|---|---|---|---|---|---|---|
| Sco_PH | WGS | Long reads | PacBio Sequel II System | 15,500 | 1,792,104 | 1,786,541 | Genome Assembly, Gap Closing, Assessment |
| Sco_PE | WGS | Short reads | HiSeq X Ten | 478 | 149,564,893 | 84,738,393 | Scaffold, Assessment |

(uncorrected *p*-distances). Furthermore, the PE and PacBio HiFi mtDNA assemblies differed from the *S. colias* mtDNA assembly available on NCBI (accession number AB488406.1 [11]), by 0.35% and 0.40% respectively (uncorrected *p*-distances). The mtDNA gene content and arrangement is as expected for most fishes and is standard for vertebrates [72], consisting of 13 protein-coding genes, 22 transfer RNA (trn), and 2 ribosomal RNA (rrn) (Figure 4).

The primary genome assembly was produced using filtered PacBio HiFi reads and the below software packages and settings. Following the above-mentioned criteria (Material and Methods: Nuclear genome assembly and assessment) the Sco_k21 assembly was selected, with both pseudo-haplotypes merged and subjected to purge_dups. Detailed statistics of Hifiasm and HiCanu genome assemblies can be consulted in Additional File 4 [17, 18]. Although the purge_dumps generated a primary and an alternative assembly, only the primary assembly was used in subsequent steps. At the same time, 4 short-read genome assemblies were performed with W2RAP software, and contigs with over 500 bp were used as "long reads" to scaffold the primary assembly. Additional File 5 shows QUAST and BUSCO statistics for the PE genome assemblies [17, 18]. Importantly, during the scaffolding process, only structural information of short-read assemblies was used, without the inclusion of bases. Lastly, the remaining non-basal long read assemblies were used to fill gaps inserted during the scaffolding stage. The final assembly (primary assembly) of *S. Colias* yielded a genome size of 814 Mbp, distributed in 2,028 scaffolds and 2,093 contigs with an N50 length of 4.19 and 3.34 Mbp, respectively. On the other hand, the alternative assembly had 807 Mbp and 5908 contigs with an N50 length of 0.47 Mbp (Table 2). The BUSCO analyses, at the nucleotide level, in Eukaryota and Actinopterygii datasets, showed high levels of completeness for both primary (97.3% and 97.9% of single-copy orthologs) and alternative (93.3% and 96% single-copy orthologs) assemblies (Table 2). Consistently, Merqury determined high QV (primary, 56.53%; alternative, 54.99%) and $k$-mer completeness (primary, 86.11%; alternative, 84.60%) values for both assemblies (Table 2). In the primary assembly, the $k$-mer analyses (via Merqury) showed a low level of $k$-mer duplication in the genome (colour blue, green, purple, and orange in Figure 5a), indicating a high level of haplotype uniqueness (red colouring in Figure 5a), and a similar $k$-mer distribution pattern to GenomeScope2 (performed with Illumina PE reads). Additionally, we found a high mapping rate in the Illumina, PacBio, and RNA-Seq reads, against the primary assembly of 95%, 99.8%, and 90.02%, respectively. Overall, these results provide evidence of the high quality of the *S. colias* genome assembly (Table 2). Our *S. colias* genome assembly ranks fourth in high-quality genome assemblies within the order Scombriformes and first in the genus *Scomber* (Additional File 6) [17, 18].

The RepeatMasker software masked 29.62% of bases in the primary genome assembly. The masked regions were predominantly linked to DNA elements (11.66%), long interspersed nuclear elements (4.11%), long terminal repeats (2.58%), and simple repeats (2.88%). Furthermore, 8.62% of the genome was masked and annotated as "unclassified", and only a small percentage were classified as short interspersed nuclear elements, small RNA, or satellite repeats (Table 3). The genome annotation process generated about 27,675 protein-coding genes and 30,999 protein-coding sequences. On average, we found 9.5 exons and 1,656 bp lengths per CDS (Table 4). Of the CDS, 30,355 had at least 1 BLASTP hit in SwissProt or RefSeq databases, 27,101 were identified in the InterPro database, and 21,664 of these were classified as belonging to a specific homolog superfamily (Table 5).

To validate the protein-coding sequences we performed phylogenetic analysis (via OrthoFinder) and BUSCO analysis (using the Actinopterygii library profile) (Figure 5b, c). Of the 16 Actinopterygii proteins datasets inputted to OrthoFinder, 98.3% were assigned to 29,066 orthogroups, with 12,334 orthogroups present in all species. All OrthoFinder statistics can be consulted in Additional File 7 [17, 18]. Furthermore, a total of 392 single-copy orthologues were retrieved by OrthoFinder and used for the phylogenomic

**Table 2.** Statistics of the *Scomber colias* genome assembly.

| Assembly | Alternative | Primary | |
|---|---|---|---|
| | Contigs (Sco_k21_a_c) | Contigs (Sco_k21_p_c) | Scaffolds (Sco_k21_p_s) |
| Number of contigs (≥10,000 bp) | 5,908 | 2,093 | 2,028 |
| Number of contigs (≥50,000 bp) | 2,417 | 1,123 | 1,078 |
| Number of contigs (≥100,000 bp) | 1,593 | 704 | 662 |
| Number of contigs (≥200,000 bp) | 1,025 | 456 | 417 |
| Number of contigs (≥500,000 bp) | 421 | 235 | 209 |
| Number of contigs (≥1,000,000 bp) | 123 | 155 | 138 |
| Total length (≥10,000 bp) | 807,928,680 | 813,976,802 | 814,072,661 |
| Total length (≥50,000 bp) | 721,244,010 | 781,696,683 | 782,480,923 |
| Total length (≥100,000 bp) | 662,374,873 | 751,893,146 | 752,912,084 |
| Total length (≥200,000 bp) | 580,469,606 | 716,806,065 | 718,068,371 |
| Total length (≥500,000 bp) | 385,329,197 | 648,055,626 | 653,890,381 |
| Total length (≥1,000,000 bp) | 180,689,595 | 591,655,104 | 603,146,189 |
| Largest contig (Mbp) | 3,248 | 22,804,600 | 22,804,600 |
| Total length (Mbp) | 807,936 | 813,977 | 814,072 |
| GC (%) | 39.94 | 40.09 | 40.09 |
| N50 (Mbp) | 0,466 | 3,342 | 4,190 |
| *K*-mer completeness (%) | 84.602 | | 86.1077 |
| Consensus quality | 56.5369 | | 54.9969 |
| Read back mapping PE (%) | - | | 95.0 |
| Read back mapping PH (%) | - | | 99.8 |
| Read back mapping RNA-Seq (%) | - | | 90.2 |
| **BUSCO statistics (databases)** | - | | |
| Eukaryota** | T: 93.3, C: 90.2 [S: 88.6, D: 1.6], F: 3.1, M: 6.7, n: 255 | T: 97.3, C: 96.1 [S: 93.7, D: 2.4], F: 1.2, M: 2.7, n: 255 | |
| Actinopterygii** | T: 96.0, C: 94.8 [S: 91.9, D: 2.9], F: 1.2, M: 4.0, n: 3640 | T: 97.9, C: 97.2 [S: 96.2, D: 1.0], F: 0.7, M: 2.1, n: 3640 | |

* Statistics are based on contigs and scaffolds of size ≥1000 bp. ** (T, total BUSCOs found (completed + fragmented), %; C, complete BUSCOs [S, complete and single copy, %; D, complete and duplicated, %]; F, fragmented, %; M, missing, %; n, number of sequences in database).

analysis. Alignment, trimming and concatenation of all single-copy orthologues, resulted in a final 120,886 aa-long supermatrix alignment that was used for phylogenomic inference in IQ-Tree. The resulting Maximum Likelihood phylogenetic tree has maximum support for almost all nodes (Figure 5b). The phylogeny recovered the reciprocal monophyletic Acanthopterygii groups Pelagiaria, Eupercaria, Anabantaria, Carangaria, and Ovalentaria, with Pelagiaria being the basal clade and represented by the 3 Scombrifomes, including *S. colias* (Figure 5b). These results are in accordance with the most recent phylogenomic study of ray-finned fishes [73], as well as the Ensembl Compara Species Tree of Ensembl database [51]. BUSCO analysis showed the *S. colias* proteome with 93.6% of the groups complete, 2% fragmented, and 4.4% missing (Figure 5c). In comparison, *T. maccoyii* had 99.8% BUSCO groups complete, while *T. orientalis* had but 82.8%. These results are expected, since the *T. maccoyii* genome assembly, part of the Vertebrate Genome Project [74], was built at chromosome level, with multiple technologies (including 46x PacBio data, 46x 10X Genomics Chromium data, BioNano data, and Arima Hi-C data) and several manual curation steps [75]. In contrast, both *T. Orientalis* [76] and *S. colias* were built at scaffold level using only short- and long-read information.

We further explored the quality of the annotation by investigating the repertoire of the NRs superfamily in the *S. colias* assembly. NRs are critical molecular physiology components, with vital roles in animal physiology and disruption [77]. Moreover, their

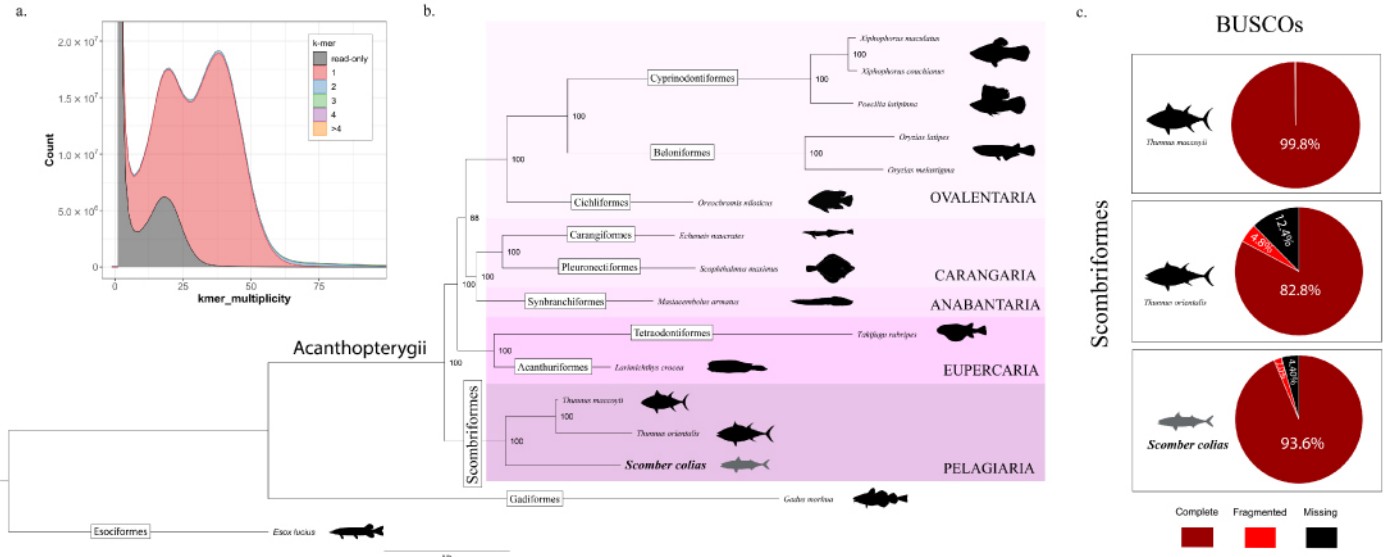

**Figure 5.** Validation of the genome assembly and annotation process. (a) *K*-mer analyses of the *Scomber colias* genome assembly (Merqury). (b) Maximum Likelihood phylogenetic tree based on the concatenated alignments of amino acid sequences of 392 single-copy orthologs retrieved by OrthoFinder. Bootstrap values are shown next to the nodes. (c) BUSCO scores were obtained from searching the proteomes of the 3 Scombriformes species with genome annotation available, against the actinopterygii_odb10 (n:3640) lineage.

**Table 3.** Report of RepeatMasker software. This report contains statistics of repetitive elements in the *Scomber colias* genome assembly.

| Total number of sequences | 2,028 | | |
|---|---|---|---|
| Total length (bp) | 814,072,661 bp | | |
| GC level (%) | 40.09 | | |
| Number of bases masked | 241,071,029 bp (29.62%) | | |
| **Type** | **Number of elements** | **Length in Genome** | **Percentage of Genome** |
| SINEs | 16,132 | 2,679,916 | 0.33 |
| ALUs | 0 | 0 | 0.00 |
| MIRs | 7,082 | 1,280,739 | 0.16 |
| LINEs | 113,089 | 33,426,533 | 4.11 |
| LINE1 | 8,048 | 4,651,362 | 0.57 |
| LINE2 | 57,670 | 14,551,177 | 1.79 |
| L3/CR1 | 697 | 123,438 | 0.02 |
| LTR elements | 82,410 | 20,969,171 | 2.58 |
| ERVL | 10 | 279 | 0.00 |
| ERVL-MaLRs | 0 | 0 | 0.00 |
| ERV_classI | 22,786 | 4,702,084 | 0.58 |
| ERV_classII 11490 | | 576,448 | 0.07 |
| DNA elements | 623,126 | 94,930,706 | 11.66 |
| hAT-Charlie | 27,952 | 5,526,534 | 0.68 |
| TcMar-Tigger | 169 | 46,619 | 0.01 |
| Unclassified | 278,199 | 70,161,089 | 8.62 |
| Total interspersed repeats | - | 222,167,415 | 27.29 |
| Small RNA | 11,380 | 1,807,250 | 0.22 |
| Satellites | 18,552 | 3,093,792 | 0.38 |
| Simple repeats | 84,014 | 23,465,814 | 2.88 |
| Low complexity | 959 | 200,769 | 0.02 |

exact NR gene complement in vertebrate lineages has been shown to vary [67]. We were able to deduce the existence of 76 NRs in the *S. colias* genome, detailed in Additional File 8,

**Table 4.** Structural annotation report of the *Scomber colias* genome assembly.

| Structural Annotation | Result |
|---|---|
| Number of genes | 27,675 |
| Number of mRNAs | 30,999 |
| Number of CDSs | 30,999 |
| Number of exons | 295,102 |
| Number of introns | 264,103 |
| Number of exon in CDS | 295,102 |
| Number of intron in CDS | 264,103 |
| Number of introns in exon | 264,103 |
| Number of introns in intron | 235,209 |
| Number gene overlapping | 71 |
| Number of single exon genes | 2,036 |
| Number of single exon mRNA | 2,105 |
| Mean mRNAs per gene | 1.1 |
| Mean CDSs per mRNA | 1.0 |
| Mean exons per mRNA | 9.5 |
| Mean introns per mRNA | 8.5 |
| Mean exons per CDS | 9.5 |
| Mean introns in CDSs per mRNA | 8.5 |
| Mean introns in exons per mRNA | 8.5 |
| Mean introns in introns per mRNA | 7.6 |
| Total gene length | 269,856,447 |
| Total mRNA length | 310,580,471 |
| Total CDS length | 51,346,678 |
| Total exon length | 51,346,678 |
| Total intron length | 259,233,793 |
| Total intron length per CDS | 259,233,793 |
| Total intron length per exon | 259,233,793 |
| Total intron length per intron | 35,919,947 |
| Mean gene length | 9,750 |
| Mean mRNA length | 10,019 |
| Mean CDS length | 1,656 |
| Mean exon length | 173 |
| Mean intron length | 981 |
| Mean intron in exon length | 981 |
| Mean intron in intron length | 152 |
| Longest gene | 242,447 |
| Longest mRNA | 242,447 |
| Longest CDS | 98,436 |
| Longest exon | 14,939 |
| Longest intron | 76,003 |
| Longest CDS piece | 14,939 |
| Shortest gene | 303 |
| Shortest mRNA | 144 |
| Shortest CDS | 18 |
| Shortest exon | 3 |
| Shortest intron | 30 |

in line with the repertoire described for other teleost species [78]. Among the retrieved NRs we found those that are key components of the "chemical defensome"—an ensemble of related and unrelated genes that protect organisms against chemical stressors, and are thus critical under anthropogenic chemical build-up and climate change scenarios—such as the xenobiotic-inducible pregnane X receptor (*pxr*, *nr1i2*) [68, 79]. Subsequent analysis, using gene names, further suggested the presence of gene annotations for the vast majority of the reported members of the teleost "chemical defensome" in *S. colias*, similarly to that

**Table 5.** Functional annotation report of *S. colias* genome assembly.

| Functional Annotation | Number |
|---|---|
| Swiss-Prot/ RefSeq | 30,355 |
| InterPro | 27,101 |
| CDD | 12,832 |
| Coils | 7,705 |
| GO | 18,643 |
| Gene3D | 22,209 |
| HAMAP | 463 |
| KEGG | 1,402 |
| MetaCyc | 1,140 |
| MobiDBlite | 16,765 |
| PIRSF | 1,755 |
| PRINTS | 7,143 |
| Pfam | 25,708 |
| PROSITE patterns | 8,082 |
| PROSITE profiles | 16,229 |
| Reactome | 7,376 |
| SFDL | 114 |
| SMART | 14,906 |
| SUPERFAMILY | 21,664 |
| TIGRFAMs | 1,427 |

described for *D. rerio* [68]. Additional BLAST searches were performed for a reduced set of genes (*fthl, gstp, hsph, maff, nme8,* and *slc21*), uncovering possible homologs for this gene subset, except for a single member of the GST family (*gstp*). The chemical defensome repertoire identified in *S. colias* species is detailed in Additional File 9 in the associated data entries [17, 18].

We additionally validated our dataset by examining the present population structure of the species, since the genome may also provide clues regarding its past demographic history [69]. One popular method to produce these inferences is the pairwise sequentially Markovian coalescent (PSMC) model, here applied to the *S. colias* final genome assembly. Since PSMC requires an estimation of the genome-wide mutation rate, and since this has never previously been produced for *S. colias,* we used the recently estimated genome-wide mutation rate of the yellowfin tuna, *T. albacares*, of $7.3 \times 10^{-9}$ mutations/site/generation [70]. The results suggest a past population expansion between 160,000–115,000 years ago, with maximum effective population size ($N_e$) of 36,000 during the end of the Mid-Pleistocene Transaction, corresponding to the Eemian (i.e., the last interglacial period) and the transition between Marine Isotope Stages (MIS) 5 and 6 (Figure 6). This population expansion is followed by an apparent decrease in the $N_e$ to around 25,000 at the beginning of the Late Pleistocene, corresponding to the beginning of the Last Glacial Period. These results, suggesting the influence of climatic changes from the Pleistocene glaciation cycles on the $N_e$, are following other recent studies on Scombriformes, such as the Pacific Sierra mackerel, *Scomberomorus sierra* [80], and the Indo-Pacific Yellowfin tuna *T. albacares* [70], as well as in other pelagic marine species such as the killer whale [81].

## REUSE POTENTIAL

This study reports the first genome assembly of Atlantic chub mackerel. *Scomber colias* is a valuable marine resource, with a high impact on the fisheries of several countries along the west coast of the Atlantic Ocean and the Mediterranean Sea. Ecologically, this species

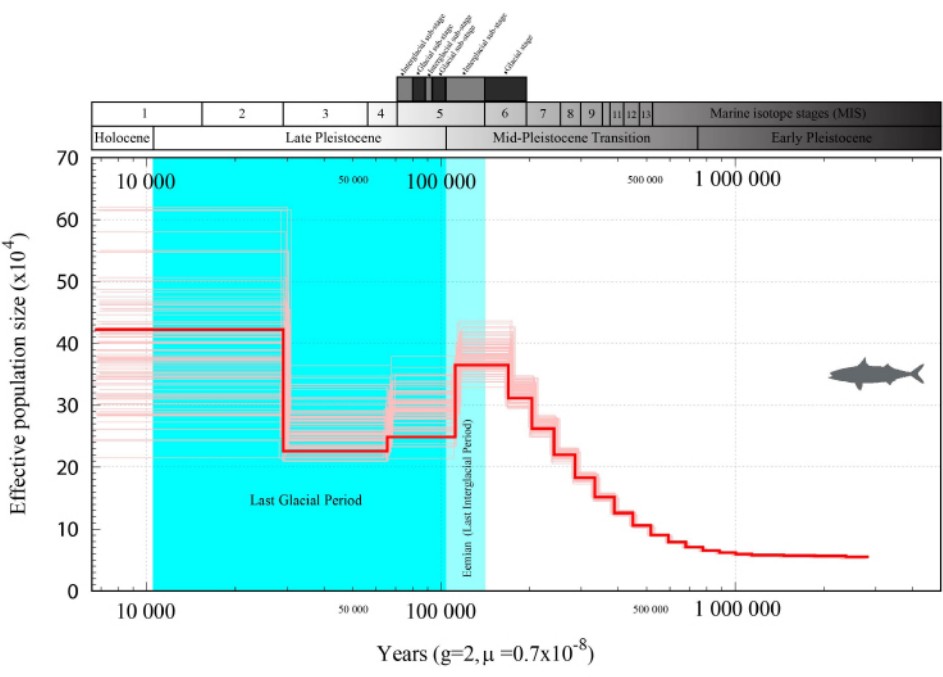

**Figure 6.** Pairwise sequentially Markovian coalescent (PSMC) estimates from the *Scomber colias* genome assembly. Estimations were obtained using a generation time of 2 years and genome-wide mutation rate of $7.3 \times 10^{-9}$ mutations/site/generation. Effective population size ($N_e$) is presented in the left vertical axis, and changes estimated up to the present, over the last 3 myr, on the horizontal axis.

establishes an important link between primary producers and top predators of the marine trophic web. Despite the ecological and economic importance of *S. colias*, few genomic resources are available for this species. Thus, this genome is timely and is expected to contribute to the effective conservation, management, and sustainable exploitation of *S. colias* species in the Anthropocene. Additionally, this genome will be a key tool to decipher biological features of *S. colias*, such as population dynamics, ecology, and physiology.

## DATA AVAILABILITY

Raw datasets of PacBio HiFi and Illumina sequencing were deposited in the NCBI Sequences Read Archive under Bioproject PRJNA769550. Additionally, both primary and alternative pseudo-haplotype assemblies were submitted to NCBI GenBank (accession numbers JAJDFG000000000 and JAJDFH000000000). Mitochondrial genome assemblies and annotations were submitted to GenBank (accession numbers OK501306 and OK501307). The four W2RAP assemblies, as well as genome annotation files and supplementary tables, were uploaded to Figshare online repository [17]. Additional data is available at the *GigaScience* GigaDB repository [18]. Genome and annotation datasets also can be interactively explored at http://portugalfishomics.ciimar.up.pt/app/scombercolias/.

## DECLARATIONS
## LIST OF ABBREVIATIONS

aa: amino acid; AGAT: Another Gtf/Gff Analysis Toolkit; bp: base pair(s); BUSCO: Benchmarking Universal Single-Copy Ortholog; BWA: Burrows-Wheeler Aligner;

CDS: protein-coding sequences; DHA: docosahexaenoic acid; gDNA: genomic DNA; HiFi: High-Fidelity; KAT: *k*-mer analyses toolkit; LINKS: long interval nucleotide *k*-mer scaffolder; Mbp: megabase pair(s); mtDNA: mitochondrial genome; NCBI: National Center for Biotechnology Information; NRs: nuclear receptors; NT-NCBI: nucleotide database of NCBI; PacBio: Pacific Biosciences; PE: paired-end; PSMC: pairwise sequentially Markovian coalescent; QUAST: Quality Assessment Tool for Genome Assemblies; QV: consensus quality value; RNA-Seq: RNA sequencing; SMRT: single molecule, real-time; SNP: single-nucleotide polymorphism.

## ETHICS APPROVAL

This work has been approved by the CIIMAR ethics committee and by CIIMAR Managing Animal Welfare Body (ORBEA) according to the European Union Directive 2010/63/EU.

## COMPETING INTERESTS

The authors declare that they have no competing interests.

## AUTHORS' CONTRIBUTIONS

LFCC designed and conceived this work; MF, RC, and NA collected the samples; AMM, AGS, EF, LFCC wrote the manuscript; AMM, AGS, MMF, FC, MS, MD, RdF, RR, AV, and LFCC coordinated and carried out the bioinformatics analyses. All authors read, revised, and approved the final manuscript.

## FUNDING

This research was funded by COMPETE 2020, Portugal 2020, and the European Union through the ERDF (grant number 031342), and by FCT through national funds (PTDC/CTA-AMB/31342/2017), and is part of the CIIMAR-lead initiative Portugal-*Fishomics*. The Foundation for Science and Technology (FCT) Portugal supported AMM (DFA/BD/8069/2020), AGS (SFRH/BD/137935/2018), AV (DL57/2016), NA (DFA/BD/6218/2020). RRdF thanks the Villum Foundation for its funding of the Center for Macroecology, Evolution, and Climate (DNRF96).

## ACKNOWLEDGEMENTS

Not applicable.

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
