## [Reviewer Report]

Comments on revised manuscriptI'm glad to see that the author has made the necessary changes. I think this manuscript is acceptable.

---

## [Reviewer Report]

Comments on revised manuscriptThe revised manuscript is improved and the responses are satisfactory. A mistake in the workflow, Id >=0.9% should be id >=0.9. No other comments to be made. I suggest accepting the manuscript.

---

## [Reviewer Report]

Reviewer name and names of any other individual's who aided in reviewer Rong HuangDo you understand and agree to our policy of having open and named reviews, and having your review included with the published papers. (If no, please inform the editor that you cannot review this manuscript.)YesIs the language of sufficient quality?YesPlease add additional comments on language quality to clarify if needed
Are all data available and do they match the descriptions in the paper? YesAdditional CommentsAre the data and metadata consistent with relevant minimum information or reporting standards? See GigaDB checklists for examples <a href="http://gigadb.org/site/guide" target="_blank">http://gigadb.org/site/guide</a>YesAdditional CommentsIs the data acquisition clear, complete and methodologically sound?YesAdditional CommentsIs there sufficient detail in the methods and data-processing steps to allow reproduction?YesAdditional CommentsIs there sufficient data validation and statistical analyses of data quality? Not my area of expertiseAdditional CommentsIs the validation suitable for this type of data?YesAdditional CommentsIs there sufficient information for others to reuse this dataset or integrate it with other data?NoAdditional CommentsIt is suggested that the author can make a simple table to show the assembly effects within the Order Scombriformes.Any Additional Overall Comments to the AuthorScomber colias is a valuable marine resource, with a high impact on the fisheries of several countries on the west coast of the Atlantic Ocean and/or the Mediterranean Sea. This study reports the first genome assembly of Atlantic chub mackerel. This genome is timely and the assembly process is clearly describedis, which contribute to the effective conservation, management, and sustainable exploitation of S. colias species in the Anthropocene. I still have the following questions.

The assembly effect of the genome does not seem to be particularly good. For example, the length of N50 length of scaffolds is not long enough. How many ploidy is this species? Do heterozygosity and repetition rate affect the assembly effect? 

It is suggested that the author can make a simple table to show the assembly effects within the Order Scombriformes. It is helpful for relevant researchers to make use of the genomic resources.

Is "data validation" followed by the results section? And there is no subtitle in the result part. Is it required by this type of article?
RecommendationMajor Revision

---

## [Reviewer Report]

Reviewer name and names of any other individual's who aided in reviewer Jianbo JianDo you understand and agree to our policy of having open and named reviews, and having your review included with the published papers. (If no, please inform the editor that you cannot review this manuscript.)YesIs the language of sufficient quality?YesPlease add additional comments on language quality to clarify if needed
Are all data available and do they match the descriptions in the paper? YesAdditional CommentsAre the data and metadata consistent with relevant minimum information or reporting standards? See GigaDB checklists for examples <a href="http://gigadb.org/site/guide" target="_blank">http://gigadb.org/site/guide</a>YesAdditional CommentsIs the data acquisition clear, complete and methodologically sound?YesAdditional CommentsIs there sufficient detail in the methods and data-processing steps to allow reproduction?YesAdditional CommentsIs there sufficient data validation and statistical analyses of data quality? YesAdditional CommentsIs the validation suitable for this type of data?YesAdditional CommentsIs there sufficient information for others to reuse this dataset or integrate it with other data?YesAdditional CommentsAny Additional Overall Comments to the AuthorThis submission described a reference genome for the Atlantic chub mackerel (Scomber colias) using the combination of PacBio HiFi long reads and Illumina short reads. The sequencing data process and genome assembling and related bioinformatics are comprehensive and adequate. The reported reference genome is the first genome and good continuity. It is a pity that the genome is not the chromosome level due to lack of the Hi-C data or genetic map data. However, the associated analysis and results make sense. In my opinion, as the first reference genome in the genus Scomber, this reference genome is a valuable genomic resource for population genetics, ecology and physiology and other future research. I have some concerns that should be addressed before publication in GigaByte.

1) In the project design, for genome assembly, two individuals were used for genomics DNA extraction. Why not used the same individual for avoiding the assembly error due to the genetic different between individuals? 
2) Line 186-196, I have some confuse about the contamination process, is there some contamination in your sample? In general, most of the genome project will not contain contamination. This process is effective for the specific sample to avoid the contamination.
3) In Phylogenomics analysis, the divergence time was recommended, then the Figure should be updated make more sense. 
4) Supp. Table 6 is blank.
5) All of the supplementary tables were not shown in manuscript. 
6) The genome assemble for Illumina sequencing is useless compared with HiFi data. 
7) In supplementary Table 5, N50 (Kb) should be N50 (bp). 
RecommendationMinor Revision